# Experiences of ‘Indirect’ Illness in Family Caregivers of Chronically Ill and Older Adults: A Qualitative Study

**DOI:** 10.3390/ijerph22020240

**Published:** 2025-02-08

**Authors:** Antonio Bonacaro, Chiara Cosentino, Concetta Collaro, Maria Chiara Gandini, Rachele La Sala, Giovanna Artioli, Victoria Cervantes Camacho

**Affiliations:** 1Department of Medicine and Surgery, University of Parma, 43121 Parma, Italy; antonio.bonacaro@unipr.it (A.B.); chiara.cosentino1@unipr.it (C.C.); concetta.collaro@unipr.it (C.C.); rachele.lasala@unipr.it (R.L.S.); victoria.cervantescamacho@unipr.it (V.C.C.); 2Social and Health Company, Grande Ospedale Metropolitano Niguarda Cà Granda, 20162 Milan, Italy; mariachiara.gandini@ospedaleniguarda.it

**Keywords:** family caregiver, caregiver–patient relationship, older adult, ‘indirect’ illness experience, chronic illness, caregiver support, qualitative research

## Abstract

The care of chronically ill and older adults affects families in terms of physical, emotional, relational, and social impact; the functioning of the family; and an alteration in family dynamics. This study aimed to evaluate the experiences of family caregivers of older patients in depth. A qualitative study was conducted, consisting of twenty-four narrative interviews with caregivers of chronically ill and older adults, consisting of twenty-three women and one man. The caregivers were all close relatives of the patients. The main results were as follows: (i) caregiving is predominantly performed by females; (ii) the caregiver experiences an ‘indirect’ illness; (iii) the family undergoes changes in roles, responsibilities, and relationships; (iv) support needs emerge, especially those of a social, family, work, and economic nature; and (v) resources are also activated, linked to the presence or absence of self-care. Illnesses directly impact the primary caregiver, but they also have repercussions on the entire family unit, which may activate coping strategies or lead to breakdown. It is crucial to accurately monitor the new family dynamics in order to find suitable solutions to support the family of chronically ill and older patients, underscoring the importance of work in this area.

## 1. Introduction

The European population is characterized by an increased percentage of older adults. On 1 January 2023, people aged 65 and over comprised 21.3% of the European population, an increase of 3% compared to the previous 10 years [1]. This correlates with an increased overall absolute disease burden [2], as 36% of people aged 65 and over in Europe reported having at least two chronic diseases on average [3]. Caring for chronically ill and older adults affects families in terms of physical, emotional, relational, and social impacts, the functioning of the family, and alterations in family dynamics, as well as the strategies implemented to cope with a prolonged illness characterized by chronicity [4].

A family caregiver is a member of the family who is intensively involved in the care of the ill person [5]. This person plays a key role in daily living activities (e.g., bathing, meal preparation), medical care (e.g., medication support, treatment decisions), household tasks (housekeeping, caring for other family members), financial administration, psychosocial care (e.g., companionship, communication with family and friends), and advocacy (e.g., when navigating through healthcare and when dealing with insurance issues) [5]. This can result in a substantial burden that could have a negative impact on young caregivers’ mental and emotional well-being and result in social isolation and lower life satisfaction [6]. A recent literature review of qualitative studies showed how being a family caregiver could have some negative outcomes, such as physical exhaustion related to caregiving, the disruption of personal life plans and psycho-emotional and socio-economic consequences [7,8]. However, some positive outcomes also emerge, such as achieving balance, a promotion in relationship intimacy, the introduction of meaning and purpose, and spiritual growth [9], all of which are thought to be strictly related to the self-efficacy and resilience of caregivers [10,11]. On the other hand, the negative impacts on psycho-emotional status seem to be strictly related to the sacrifices and adjustments that caregivers are forced to make to adequately fulfill their caregiving role. Changes in work, leisure activities, and peer activities significantly impact depression levels in older caregivers [12,13].

The whole experience of caring for a chronically ill or older person can completely alter pre-existing family dynamics. A qualitative meta-synthesis showed that caring for people with dementia can impact a family in four main ways: the foundation of the family can crack, marked boundaries of care may be voluntarily or involuntarily setup, the role of the family may change (a supportive environment versus a reluctant or interfering environment), and relationships can be re-established within and outside the family [14]. Communication within the whole family can be altered with experiences of denial and inability to understand the caregiver’s challenges [15]. Moreover, within the family dynamics, and especially within caregivers, guilt can frequently be experienced [16], as there is a sense of inadequacy regarding the care provided to the recipient. Specifically, a qualitative study unveiled four main areas of guilt in cancer caregivers: guilt due to blaming the patient for the illness, guilt due to a lack of self-capacity, guilt for neglecting others, and no guilt at all [17]. Another qualitative study underlined seven areas of guilt in dementia caregivers: guilt derived from actions themselves, guilt derived from one’s limitations, guilt for feeling negative emotions, guilt associated with the change in the relationship with the person receiving care, guilt for neglecting other areas, guilt induced by the person receiving care, and guilt induced by others [18].

Some specific differences emerge depending on whether the family caregiver is male or female. Two-thirds of caregivers are women [19] and, despite no differences in the type of caregiving (except for household chores) and the reported amount of time spent on it [20], they seem prone to developing higher levels of stress, anxiety, and depression [21]. A recent systematic review confirmed these data, linking them to over-responsibility and self-sacrifice, common in female caregivers [22]. Moreover, for female caregivers specifically, the impact of burnout worsens over time; this is especially linked to poorer patient health and a decrease in fulfillment [23].

Although the literature reveals the wide impact that caregiving has on a family system and on the life of the caregiver, especially when the caregiver is a woman, less emerges about the type of experience that is lived and, especially, how they live it, which brings a unique perspective to the caregiver role.

This study aimed to understand, in depth, the experiences of family caregivers of older patients.

## 2. Materials and Methods

### 2.1. Study Design

This study was based on a descriptive phenomenological methodology. In the phenomenological tradition, the thematization of meaning can be understood as being related to the interpretation of data, illuminating underlying, tacit or hidden meanings in participants’ lived experiences [24,25]. In descriptive phenomenology, thematic analysis focuses on the evidence; it is placed in the human experience. In this study, we used a thematic analysis based on the approach of Braun and Clarke. Thematic analysis was carried out following 6 steps [26]: 1. Familiarization with the data, reading and re-reading the texts and noting initial ideas. 2. Generation of the initial coding, as well as identification of the first units of meaning. 3. An initial search for themes that emerged from the data, defining sub-themes and themes. 4. A review of topics with the research team. 5. The definition, naming, and renaming of topics. 6. Report production. To ensure the scientific nature of this analysis, we validated the emerging themes, first independently and then together. This study followed the consolidated criteria for reporting qualitative research (COREQ) checklist for interviews [22].

This study received authorization from the Research Ethics Board (REB) of the University of Parma, with protocol number 0198268, on 18 July 2023.

### 2.2. Participants and Setting

Convenience sampling was carried out among the relatives of patients referred to hospital and home healthcare service facilities in Northern Italy, settings in which some members of the research team were employed. Members of the research team (M.C.G. and C.C. (Concetta Collaro)) contacted all the participants in person and invited them to an online interview via dedicated platforms. M.C.G. and C.C. (Concetta Collaro) identified possible participants based on the inclusion criteria described below, informed the identified caregivers in detail about the study and participation in the interview, handed over the information and consent form, and explained that the written consent would be handed over to the ward coordinator, to ensure free consent. M.C.G. and C.C. (Concetta Collaro) work in the clinic and service where the participants were recruited. All contacted caregivers agreed to participate in the interview.

The inclusion criteria for the selection of the participants were as follows: caregivers of patients over 65 years old with no cognitive disorders that prevent them from understanding the questions and their answers given to them; proficiency in the Italian language; and that they were continuously in the presence of their relative (more than 12 h a day). The following were excluded: caregivers who were unable to answer the interview questions; caregivers of another nationality who were unable to understand and speak Italian; and family members who were not the main caregiver.

The initial number of participants in the study was 18. It was thus necessary to redefine the sample to reach data saturation by including six participants [27]. When recruiting participants, the principles of pluralism, protection of minorities, and non-discrimination were observed. The subjects invited to participate in the study did not find themselves in a position of dependency vis à vis the researcher, so the expression of consent to participate was entirely free and free from any kind of pressure.

### 2.3. Research Team

The research team consisted of people of multiple professions: two PhD psychologists (C.C. (Chiara Cosentino) and V.C.C.), two registered nurses (M.C.G. and C.C. (Concetta Collaro)), one expert in education with an MSc in Nursing (G.A.), and two PhD RNs (R.L.S. and A.B.). All researchers had previous training in thematic analysis and experience in qualitative approaches. There were six females and one male in the research group, and all researchers worked at universities as researchers and adjunct professors. They had no previous dealings with the study participants, except for the two members who first contacted the participants (M.C.G. and C.C. (Concetta Collaro)).

### 2.4. Data Collection

A semi-structured narrative interview was used for data collection. This research technique was useful because the aims of this study were to investigate complex or unfamiliar topics, to collect data from real-life situations, and to answer open-ended research questions that were not limited by a structured questionnaire [28]. Narrative interviewing also promoted the participant’s verbal expression as the interviewer asked open-ended questions, left plenty of room for answers, and listened to answers in a non-judgmental manner [29].

All the interviews were conducted online via web platforms, with audio/video recording with the participant’s consent. Family caregivers were asked to be available to be contacted afterwards in the event of the need for more information.

The interview questions were expanded collectively in groups so that they were as open and non-directive as possible, neutral, and conducive to reflection.

The interview was structured into 4 sections, each with at least 1 question, designed to elicit information relevant to the study’s objective. Table 1 provides a detailed overview of the interview topics and the main questions to be asked.

We tested the interview guide and made no changes.

The interviews lasted between 20 and 40 min.

While conducting the interview online, the interviewer ensured that the caregiver had a suitable setting to guarantee the person’s privacy: a silent environment without distractions.

The interviewer actively ensured that the interview corresponded to the cognitive purposes proposed by the study, creating a climate of trust and listening. This approach favored the active participation of the interviewee and the extraction of meaningful information.

The interviewer was attentive to the interviewee’s answers using mainly para-verbal and non-verbal communication. Based on the information during the interview, the researcher would elaborate on certain aspects or ask for clarification. An observer was present at each interview, mainly writing field notes on the interviewee.

The observers noted every element of the interviewee’s verbal and non-verbal communication and generated field notes (FNs) that would supplement the interview transcription.

The interviewers were two members of the research group (G.A. and V.C.C), experts in qualitative interviews, while the observers were C.C. (Chiara Cosentino), C.C. (Concetta Collaro), M.C.G., and R.L.S.

Once the interviews had taken place, they were coded with a progressive number to ensure anonymity, and a table was constructed containing primary biographical data.

The interviews were conducted between September 2023 and November 2023.

### 2.5. Data Analysis

The interviews were manually transcribed ’word for word’ in their entirety by the interviewers themselves and observers. This allowed for annotation of the field notes or assessment of the interviewees based on the registration data and videos. The research team did not use any analysis software to insert codes, only an Excel program.

The analysis of the audio/video recordings made it possible to focus not only on the verbal content but also on all aspects of non-verbal communication (body movements, proxemics, gestures, para-verbal communication, mimicry), and, thus, all aspects relating to the emotional experience (field notes) that emerged during the interview. [30].

The interviews were analyzed to capture the deeper meaning of the study participants’ experiences. The researchers inductively analyzed the data and manually created tree structures to move from meaning units to sub-themes and central themes.

Following the methodology of Braun and Clarke [31], the interviews were read several times by each researcher to obtain an overall understanding of the text.

In pairs and independently, four researchers (C.C. (Concetta Collaro) and M.C.G.; R.L.S. and C.C. (Chiara Cosentino)) analyzed the interviews, first defining the codes (or verbatim) and then the various units of meaning. These were condensed while retaining as many of the interviewee’s expressions as possible and then coded and entered into an analysis grid that was shared within the research team and discussed. At this stage, the researchers exchanged views to agree on the first analyses carried out. In the case of disagreement, we sought the valuable input of experienced researchers (G.A. and V.C.C.) to ensure the credibility of our findings.

In this process, the researchers also considered the field notes written by the observers and identified other field notes that emerged from the reflection on the results by the entire research team.

All meaning units were then grouped, assessing whether some common themes/essences could be merged into common meanings. This enabled the identification of sub-themes and central themes.

In the first 18 interviews, 1125 meaning units were identified. This allowed for an initial identification of the main themes that emerged. We created an initial codebook for thematic analysis into which we inserted the first twenty interviews. We then conducted and analyzed further interviews (six) to make sure there was no new information before concluding that the data were saturated [32,33].

At this second stage, 330 items of interest were identified.

The themes and sub-themes that emerged were compared in the group and were reworked and renamed to improve their adherence to the data.

### 2.6. Rigor

The research team tried to adhere to the following criteria: reflexivity, credibility, and transferability, for determining the scientific rigor of qualitative research. Reflexivity was maintained throughout the analysis, with the entire research team also keeping a reflective attitude. The analysis was transparent, which means that the researchers have presented it as comprehensively as possible to retain credibility. Transferability refers to the usefulness and relevance of results. The relevance, usefulness, and significance of the research results for other contexts are essential components of the transferability of the study [33].

## 3. Results

### 3.1. Participant Characteristics

A total of 24 interviews were carried out with primary family carers, i.e., carers who are continuously around their relatives. Twenty-three interviewees were female, and one was male. The patients’ average age was 67.9 years, with an SD of 5.94. Regarding the nature of their relationship with carers, nine respondents were daughters, nine were wives/partners, four were sisters, one was a daughter-in-law, and one was a brother. The patients were affected by the following illnesses: oncological, chronic degenerative, and mainly neurological diseases. The average duration of the interviews was 35 min.

### 3.2. Main Themes

Five key themes emerged from the data analysis: (i) caregiving is a female role; (ii) the caregivers experience an ’indirect’ illness; (iii) the family changes roles, responsibilities, and relationships; (iv) support needs emerge, especially of a social, family, work, and economic nature; and (v) resources are also activated, linked to the capacity for self-care, which become limited when there is a lack of self-care. The themes and their sub-themes are described in Table 2.

#### 3.2.1. Caregiving Is a Female Role

This theme emerged from the characteristics of the almost exclusively female sample and the interviewers’ and observers’ field notes (FNs).

From the point of view of responsibility, in most of the interviews analyzed, it was possible to distinguish a recurring pattern in which, in the presence of a sick family member, it was “the woman who tended to take on more responsibility for her relative” (FN 1a) and who was “defined by the family as the main caregiver” (FN 4).

This research showed that “women assumed the majority of caring responsibilities” (FN 3), “reflecting traditional gender roles in which women were more likely to perform domestic and caring tasks” (FN 2a). This theme was presented in the following two sub-themes:*(i)* *Disproportionate emotional and physical burden on women*

Women caregivers of older relatives “frequently suffered from social isolation” (FN 2b), as their constant dedication to caring reduced their time available for social and recreational activities. This aspect led “many of them to feel lonely and unsupported, with a negative impact on their mental well-being and quality of life” (FN 3). In addition, “pro-longed caregiving could have serious consequences for women’s physical health” (FN 1b). Chronic stress, combined with a lack of time to care for themselves, could increase the risk of developing depression, anxiety, and physical problems such as chronic pain or cardiovascular disorders. “These factors combined accentuated the emotional and physical burden women bear as caregivers.” (FN 5a).

*(ii)* 
*Professional and Economic Consequences of Female Caregiving*


Women who were primary caregivers for family members with chronic illnesses experienced “a significant economic impact as they often reduced paid work hours” (FN 5b), took part-time jobs, or left the labor market, and “faced occupational penalties due to the need to balance work and caregiving” (FN 6), which limited career advancement opportunities and increased the risk of occupational stagnation.

#### 3.2.2. The Caregivers Experience an ‘Indirect’ Illness

When a person with a chronic/oncological disease was present within a family unit, the whole family was affected and, in a certain sense, became “sick”. The elements that most affected the family carers’ experience were the time of illness, its duration, and its speed of progression. In this second theme, three sub-themes stood out:*(i)* *A journey made together*

Caregivers described the illness as a “*journey*” (15.19) which they set out on together, often marked by moments “*of pain, despair and weariness*” (3.6), but also by “*acts of care, love, dedication*” (8.10). In some cases, there was a constant emotional impact and sharing of emotions, so much so that the patient and caregiver experienced “*the same fears, the same labors, the same apprehensions…*” (9.37). The care narratives collected from the interviews revealed aspects that often remain in the shadows: “*[I stayed by his side] right up to the end, and I could not bathe him anymore because it had become that I could not move him*” (24.29).

*(ii)* 
*The close link between sweet memories and some regrets*


The speed of deterioration left people overwhelmed and disoriented, sometimes depriving the forming dyad (the patient and the caregiver) of meaningful experiences, which left a feeling of regret: “*It was swift; we did not even have time to go for a ride to the beach anymore*” (1.14). In other cases, the family member brought back a “*sweet*” memory of their last holiday: “*The summer before I had taken him to the island of Giglio because the doctor had told me: It will be the last holiday he can take… moving around*” (10.16).

*(iii)* 
*Awareness of the caregiver*


Family members also experienced their own path of awareness, first of all with regard to their loved one’s illness, but also with regard to its impact on daily life in terms of care and changes in family life. In some cases, change was interpreted as imposed (“*it is difficult to accept the illness, especially having to change for this reason*” (5.7)), and in others, resilience prevailed and the caregiver understood that the situation must be managed and that there was no need “*to feel sorry for oneself or to despair*” (11.10) and one had to move on.

#### 3.2.3. The Family Changes

We found that the diagnosis of a chronic/oncological disease not only changed the life of the individual who fell ill but affected and changed the everyday life of the whole family, transforming the family’s identity. Three sub-themes were identified:*(i)* *The impact of the disease on family members*

The impact of the illness on the family and the caregiver, in particular, was closely linked to the evolution of the illness. There was a practical and organizational impact (“*there was a person to take care of*” (5.8)) and spaces in the home were reorganized to “*make room for the medical equipment*” (8.10) needed to manage the sick person. The economic impact was also felt: “*the nappies they give are not enough*” (6.43) and “*the care allowance, which is fine*” (1.20) only covered one-third of the care costs. When the illness went on for a long time, significant emotional repercussions emerged in the caregiver, to the point where it manifested physical and emotional exhaustion of the family, well described in this quote:

“Both my daughter and I were exhausted. (…). There were times when I would bang my head against the cupboard, I would go into crisis, I could not understand him, I could not understand his needs anymore because he did not talk anymore” (1.31).

With the increase in care duration, in addition to the physical commitment, the role of support for the sick person and the other relatives became increasingly critical, with consequent significant emotional fatigue; as one interviewee said, they were tired of “*putting and always having a smile on my face*” (3.78).

*(ii)* 
*The transformation of roles*


Most of the interviews highlighted the change in roles within the family. The illness required a reorganization of the family context. Sometimes, the illness has led to a reversal of roles; as one daughter stated: “*I have become my mum’s mum*” (9.27). These significant changes led to a considerable increase in caregivers’ responsibilities and in care-related stress. The need to be close to the sick person led to some caregivers moving “*totally to live with mum*” (5.18).

*(iii)* 
*Relationships within the family*


In the interviews analyzed, almost all families were united in caring for the sick person. People rallied around the patient and intensified their interpersonal relationships in order to cope with the difficult situation: “*Things between us have changed in the sense that we have tightened up more, we have united to cope with the situation and to support dad (…)*” (2.26). In some cases, the caregivers tended to enter a process whereby they did not have to ask anything of anyone and therefore tried to organize themselves to manage everything autonomously: “*I’ve always had that thing of having to protect someone*” (20.28). Family cohesion was demonstrated in the common choices made and in the improvement of communication within the family unit. Only in a few cases did the illness of a family member generate conflicts between family members.

#### 3.2.4. Family Support Needs

The reorganization of the family structure led to interaction with different support systems outside the family context, including health, social, and friendship networks; extended families; and work and school environments. The theme of “*support*” emerged in most of the interviews, bringing out strengths and weaknesses among families, friends and professional figures. This theme was divided into two sub-themes:*(i)* *External supports*

Social support proved to be an essential glue for the families’ stability, which was undermined by the illness; friends “*were always there*” (3.68) and neighbors became so familiar that the sick person called them “*sons and grandchildren*” (17.33). Work support turned out to be a key element for greater balance in living closer to their loved ones during their illness for many interviewees, from “*much easier shifts*” (17.32) to “*hourly flexibility, maximum freedom to manage me*” (1.35). Associative support was also helpful, e.g., the “*family and Alzheimer’s group, which provided staff ’to stay with them [the patients] free of charge’*” (6.6).

*(ii)* 
*Relations with professionals*


In the various interviews, strengths and weaknesses in the relationships between the interviewed family members and health professionals emerged. The “*closeness*” of the health professionals to the family was felt, to the extent that participants said that “*People who put themselves in the place of […] who are part of the family*” (4.80). The healthcare team became a “*point of reference*” (13.20) and, by making the sick person feet good, indirectly, “*it also makes the family feel good (…)*” (2.51). When a strong relationship of trust was established, the family member felt “*in a barrel of iron*” (15.38), providing great reassurance.

There was no lack of relationship difficulties, understood as “*coldness*” in interpersonal relations (12.28), or doubtful recognition of the competence needed “*to assist such a complex person*” (1.23). An aspect eliciting very strong opinions was the direct communication of health professionals with patients about their condition; in several interventions, family members expressed the need for “*delicacy*” and “*sensitivity*” (18.17) on the part of health professionals, arguing the right to “*hear first from a family member and then, with the right tact tell them…*” (14.35) to determine whether or not the sick person was ready to listen.

Several interviewees reported how the psychological support of the family unit was incisive (“*We all started a path together with a psychotherapist*” (17.20)) for them to be able to process and face the new experience of illness.

#### 3.2.5. Caregivers’ Resources

Interviewees described that caring for a family member with a chronic or progressive disease often involved having to sacrifice one’s physical, mental, and emotional well-being. Resources were also utilized, linked to the capacity for self-care, which became limited when there was a lack of self-care. Two sub-themes were identified in this theme:*(i)* *Caring for oneself and the sick person*

Among the activities that family members undertook while caring for the sick person were sporting activities: “*I used to take my volleyball bag and go*” (20.45) and “*My mother joined walking groups, and now you see her walking in the mountains*” (17.48). These activities were undertaken to find an environment that would allow them “*to unplug a bit mentally*” (11.46).

There was an awareness that dedicating oneself to one’s family by sharing moments of leisure and relaxation was essential to preserve the well-being within one’s household: “*You have to detach yourself from her for a moment because your family needs you because you need her mentally*” (5.42).

In some cases, it was considered essential for the carer to be able to balance their well-being with the needs of the sick person: “*You are close to him, lying down next to him… that is enough, even if you play with the tablet because it is true, I used to play with the tablet just not to think*” (15.48) or “*in the last period […] I used to do the crosswords because they helped me to take my mind off it totally…*” (9.42).

*(ii)* 
*Caregivers’ self-care and lack of self-care*


During the course of their family member’s illness, a self-care-conscious approach helped support the caregivers’ resilience and adaptability in the face of difficulties. In other cases, caregivers recognized that “*I just forgot about me*” (24.34) and “*it was not possible in those 13 years ever, even one weekend, even just going out one evening or half a day, one Sunday*” (6.55). The ability to accompany a loved one on their journey was an act of profound respect and dedication that significantly transformed those who experienced it: “*You annihilate yourself as a person because you dedicate yourself completely to the person in need. Always*” (21.45). On such difficult journeys, caregivers experienced a reduced ability to take care of themselves.

## 4. Discussion

This study aimed to understand the experiences of family caregivers of older patients with chronic illnesses. Several key themes emerged from the analysis of interviews and field notes, offering valuable insights into the caregiving experience.

### 4.1. Gender and Caregiving Roles

The majority of caregivers in this study were women, a finding consistent with those in the existing literature, which indicates that two out of three caregivers are women [19]. This gender disparity can be attributed to historical societal roles, in which women have traditionally taken on family responsibilities due to their lower involvement in the labor market [34,35]. While recent decades have seen increased numbers of women in the workforce and the greater involvement of men in caregiving, women still bear a disproportionate burden of caregiving tasks [20,36]. This gender imbalance has significant implications, as female caregivers report higher levels of stress, depression, anxiety, and social tension compared to their male counterparts [21].

The field notes and interviews in this study further confirmed the mental and emotional toll on caregivers, particularly among women, and highlighted their experiences of self-sacrifice and personal strain [21,22,23].

### 4.2. Indirect Experience of Illness

A notable finding from this study is the “indirect experience of illness” that caregivers endure. Similarly to patients, caregivers experience the emotional stages of chronic illness, such as denial, anger, bargaining, depression, and acceptance [37].

This empathic connection synchronizes emotions between the caregiver and the patient, intensifying the caregiver’s emotional burden [38,39].

Caregivers not only manage the logistical challenges of care, but also share in the patient’s emotional journey, experiencing complex feelings such as guilt, anger, and uncertainty [18,40]. This profound emotional connection underscores the need to support caregivers in regulating their emotions and in developing effective coping strategies to navigate these challenges [40], focusing on the ability of caregivers to regulate their emotions and develop their coping skills.

### 4.3. Caregiver Resources and Adaptation

This study also emphasized the dual nature of caregiver resources. Some caregivers reported sacrificing their own interests and well-being to focus entirely on the patient, often leading to exhaustion and the depletion of inner resources. This phenomenon appears to be linked to the cultural values of self-sacrifice, particularly in Italian culture, and disproportionately affects women [22].

Conversely, other caregivers demonstrated resilience by practicing self-care, engaging in activities such as exercise, or taking time for themselves, which helped them to preserve their internal resources and manage caregiving challenges effectively [38]. These findings highlight the importance of promoting self-care and resilience-building among caregivers.

### 4.4. Family Dynamics and Support Systems

The participants reported significant disruptions in family roles and routines due to caregiving, with primary caregivers often taking on responsibilities akin to “becoming their parent’s parent.” This role reversal increased stress, responsibility, and emotional involvement [41,42]. Beyond the primary caregiver, the entire family experienced changes in roles and relationships, necessitating support for all members.

Previous research [43] suggests that families with a strong internal adaptability can better manage the challenges of chronic illness. Additionally, recent systematic reviews emphasize the buffering effect of social support and professional guidance in mitigating the social impact of illness on patients, caregivers, and their families [44].

These findings underline the importance of addressing family systems as a unit of analysis and intervention in caregiving contexts.

### 4.5. Limitations

Although this study provides important information on the experience of caregivers of older patients with chronic diseases, it has some limitations that must be considered when interpreting the data.

First, the research sample consisted of 23 female caregivers and only 1 male caregiver. The women were all white Italians with access to public health services. This limited the representativeness of the results in other socio-cultural contexts or for caregivers with different characteristics, such as women of different ethnic origins, male caregivers, or those who do not have easy access to health services. The predominance of participants with a good command of the Italian language also excluded the experiences of those who might have encountered additional language barriers in the context of care. However, this study is transferable to other contexts, as we explored the depth of a wide range of psychosocial processes. Other studies could, based on these data, consider the same phenomenon in different contexts and populations.

Secondly, the research was carried out in specific regions of Italy and did not cover the whole country. Although the Italian Health Service is national, over time, regional autonomy has differentiated services for different people. This may have limited the representativeness of experiences, especially in relation to regions with different levels of access to health and social support services. The influence of regional differences has not been considered in this analysis, which may have an impact on the variability of caregiving experiences. Again, the data collected and analyzed could be the basis for future studies.

### 4.6. Implications for Policy and Practice

Based on our results, the authors suggest some guidelines for clinical practice.

The patient–family dyad: It is necessary to consider the family caregiver as part of the dyad being cared for. In this way, the attention of the care team would naturally be directed towards both the patient and the family member, with the possibility of highlighting caregiver problems at an early stage.

Attention to the whole family: In a historical and social period in which the family has substantially changed with regard to its constitution and role, it is essential to direct the attention of caregivers and psychosocial services to the dynamics that develop within the current family when one of its members falls ill. This early attention to the family, with fragilities in its internal dynamics, could help to identify adequate support.

Family meetings: Family carers consider the relationship with the care team as supportive. This may favor, in practice, formal meetings between patients/family carers and professionals and family meetings, where information and concerns can be shared and support from competent staff can be offered.

Psychological support: It is essential to identify the burden of caregivers at an early stage, especially in those who already have a diminished capacity for self-care, in order to encourage them to accept the help of specialized personnel such as psychologists. Administrations should also recognize the need for expert professionals within care teams, especially for older chronic or oncological patients.

## 5. Conclusions

Diseases directly impact family caregivers, who care for a patient continuously over a long duration. In this study, almost exclusively women cared for their relatives. The research shows that the caregiver also experiences ‘an indirect illness’, following the patient’s illness step by step. Repercussions of this ‘indirect illness’ are observed in the primary family caregiver and also in the entire family unit. Both caregivers and the family may activate coping strategies or experience difficult situations. Role changes within the family are evident, and roles may be reversed with regard to responsibilities and relationships. The family caregiver shows a significant need for support, both in relational (within and outside the family) and social, work, or economic aspects. Caregivers also showed the ability to activate resources, especially their own, by activating self-care paths. This study also revealed a possible lack of self-care on the part of caregivers, a condition that can endanger the person’s psychological and physical health.

In conclusion, this study reinforces the need for holistic approaches to caregiver support, emphasizing gender-sensitive interventions, resources to build resilience, and strategies to address the broader impact of caregiving on family systems. Such efforts are essential to mitigate the physical, emotional, and social toll of caregiving for older patients with chronic illnesses.

## Figures and Tables

**Table 1 ijerph-22-00240-t001:** Interview topics and questions.

Interview Topic	Example Question
Caregiver’s approach to illness	Would you like to tell me about your experience during your loved one’s illness?
2. Changes within family	In your opinion, what has happened within the family? Could you give an example?
3. Needs of caregiver and family	During the course of your loved one’s illness, what were your and your family’s emerging needs?
4. Internal and external resources for caregivers	What, in your opinion, were the elements that helped you? Could you give an example?

**Table 2 ijerph-22-00240-t002:** Themes and related sub-themes.

Theme	Sub-Themes
Caregiving is a female role	1.1. Disproportionate emotional and physical burden on women
	1.2. Professional and economic consequences of female caregiving
2. The caregivers experience an ‘indirect’ illness.	2.1. A journey made together
	2.2. The close link between sweet memories and some regrets
	2.3. Awareness of the caregiver
3. The family changes	3.1. The impact of the disease on family members
	3.2. The transformation of roles
	3.3. Relationships within the family
4. Family support needs	4.1. External supports
	4.2. Relations with professionals
5. Caregivers’ resources	5.1. Caring for oneself and the sick person
	5.2. Caregivers’ self-care and lack of selfcare

## Data Availability

The original data presented in the study are openly available from Mendeley Data at DOI: 10.17632/wwkk7srgvp.1.

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
