# Peer review of "Experiences of ‘Indirect’ Illness in Family Caregivers of Chronically Ill and Older Adults: A Qualitative Study"

_ijerph, 2025, doi:10.3390/ijerph22020240_

Round 1
Reviewer 1 Report
Comments and Suggestions for Authors
Improvements should be made to the graphic aspect of the text. Some paragraphs should be removed, for example in the date analysis. The table of themes and sub-themes could also be improved in appearance and made smaller.
Author Response
For research article: 'Indirect' illness experiences of family caregivers of Chronic and Old Adults: a qualitative study.
ijerph-3269989
Response to Reviewer 1 - Comments
|
||
1. Summary |
|
|
Thank you very much for taking the time to review this manuscript. Please find the detailed responses below and the corresponding revisions/corrections highlighted in red colour in the re-submitted files.
|
||
2. Questions for General Evaluation |
Reviewer’s Evaluation |
Response and Revisions |
Does the introduction provide sufficient background and include all relevant references? |
Yes |
Thank you |
Are all the cited references relevant to the research? |
Yes |
Thank you |
Is the research design appropriate? |
Yes |
Thank you |
Are the methods adequately described? |
Yes |
Thank you |
Are the results clearly presented? |
Yes |
Thank you |
Are the conclusions supported by the results?
|
Yes
|
Thank you |
3. Point-by-point response to Comments and Suggestions for Authors |
||
Comments 1: Improvements should be made to the graphic aspect of the text. Some paragraphs should be removed, for example in the date analysis. The table of themes and sub-themes could also be improved in appearance and made smaller.
|
||
Response 1: Thank you for pointing this out. We agree with this comment. We reformatted tab 1 and table 2 and removed the paragraphs (i), (ii), (iii) in the text of the results (date analysis), while retaining the title of the pharagraphs.
|
||
4. Response to Comments on the Quality of English Language |
||
Point 1: (x) The quality of English does not limit my understanding of the research. |
||
Response 1: Thank you very much. |
Reviewer 2 Report
Comments and Suggestions for Authors
Thank you for the opportunity to review the manuscript. This topic is needed for caregiving literature. However there are some issues and concerns I have.
I do not understand the quotes that are used - with the Field note reference. I get that these are from the research notes, but there was no discussion that field notes were analyzed in the methods/data analysis section. There needs to be complete transparency with who did the interviews, who took field notes, etc.
Also - methodology - The recruitment is not clear to me. I understand recruitment occurred in clinical settings (it seems) but it says members of the team were associated with recruitment, but no names or initials were listed. In action, when you talked about consent you said consent could not be free of undue influence related to the recruitment process. But nothing was mentioned about how this was controlled for, or what was done to ensure free consent.
These are critical clarifications in method that must be shared.
Per CORG checklist, statements about the research teams reflection and connection to the why of the project are important. While there was a sharing of who the team was professionally, each members role (beyond data analysis) was not shared - interview, field note taker, etc. These roles are needed...in addition were any of those on the team providing care to any of the patient's to the caregivers identified?

Please refer to the attached article that has all my thoughts and additional grammatical notes.
Author Response
Please see the file in attachment

Reviewer 3 Report
Comments and Suggestions for Authors
Thank you for this paper. I must admit when I first read the title I thought "Not another paper that looks at family caregivers' experiences." Honestly, I think it is time to move beyond looking at experiences and figure out how to improve CARE for caregivers-- essentially build better systems to support family caregivers. While your findings were not novel, the research is well done and your findings are comprehensive.
The article is well written. I have some small recommendations. On page 2 line 46-- is there a better word than important in "Important load"?
Line 75 "Male on female" perhaps or rather than on
Study design Thematic analysis is a method not a qualitative methodology. Please see any number of books on qualitative methodolgies e.g., 1. Creswell, J.W.; Creswell, J.D. Research design: Qualitative, quantitative, and mixed methods approaches 5th ed.; Sage Publications Los Angeles, CA, 2018.
2. Morse, J.M. Qualitative research is not a modification of quantitative research. Qualitative Health Research 2005, 15, 1003-1005, doi:10.1177/1049732305280771.
3. Morse, J.M. What is qualitative research? Qualitative Health Research 2005, 15, 859-860, doi:10.1177/1049732305279135.
This Wikipedia reference summarizes:
Thematic analysis is one of the most common forms of analysis within qualitative research.[1][2] It emphasizes identifying, analysing and interpreting patterns of meaning (or "themes") within qualitative data.[1] Thematic analysis is often understood as a method or technique in contrast to most other qualitative analytic approaches – such as grounded theory, discourse analysis, narrative analysis and interpretative phenomenological analysis – which can be described as methodologies or theoretically informed frameworks for research (they specify guiding theory, appropriate research questions and methods of data collection, as well as procedures for conducting analysis). Thematic analysis is best thought of as an umbrella term for a variety of different approaches, rather than a singular method.
https://en.wikipedia.org/wiki/Thematic_analysis#:~:text=Thematic%20analysis%20is%20often%20understood,or%20theoretically%20informed%20frameworks%20for
Ethics-- Was there ethics for this study?
Limitations: Second point-- regional. Without any understanding of the Italian context, Is healthcare regional e.g., provincial as in Canada or state controlled as in the US?
I would have appreciated a bit more discussion of how ALL of your findings relate to the extant literature and whether/ how they were novel.
Your implications for policy and practice are relevant. As you note, the care trajectories are often quite long. In Canada, it is an average of 4.6 years. Are family meetings enough?
Is psychological support enough? Typically as you stated earlier anxiety and distress rise sharply when caregiving hours increase. In Canada is is 20 hours a week or more e.g.,
Pauley, T.; Chang, B.W.; Wojtak, A.; Seddon, G.; Hirdes, J. Predictors of Caregiver Distress in the Community Setting Using the Home Care Version of the Resident Assessment Instrument. Professional case management 2018, 23, 60-69, doi:10.1097/NCM.0000000000000245.
Overall, your work adds to the body of work about Caregivers' experiences. Yes, health care providers need to be trained and mandated to move past patient-focused care to dyadic care.
Author Response
Please, see the file in attachment

Round 2
Reviewer 2 Report
Comments and Suggestions for Authors
Please the document pdf for my comments. The revised version is much improved, but there are areas within the new text that lack citations, that need to be added. Thank you for continuing to improve this work, as I feel it is an improtant topic.

Author Response
Please see the file in the attachment

Reviewer 3 Report
Comments and Suggestions for Authors
Thank you for the improvements in your paper. The new discussion section is much improved. I think the readability can be further improved see the suggestions below. Then, you may want to go through and review the entire paper to see if you can improve the readability overall. I always find that when you have made changes based on reviewers' suggestions, the revisions and new additions don't flow as I want them.
Discussion
This study aimed to explore the experiences of family caregivers of elderly patients with chronic illnesses. Several key themes emerged from the analysis of interviews and field notes, offering valuable insights into the caregiving experience.
Gender and Caregiving Roles
The majority of caregivers in this study were women, a finding consistent with existing literature, which indicates that two out of three caregivers are women [17]. This gender disparity can be attributed to historical societal roles where women have traditionally taken on family responsibilities due to lower labor market involvement. While recent decades have seen increased workforce participation by women and greater involvement of men in caregiving, women still bear a disproportionate burden of caregiving tasks [18]. This gender imbalance has significant implications, as female caregivers report higher levels of stress, depression, anxiety, and social tension compared to their male counterparts [19]. The field notes and interviews in this study further confirmed the mental and emotional toll on caregivers, particularly among women, and highlighted their experiences of self-sacrifice and personal strain [19–21].
Indirect Experience of Illness
A notable finding from this study is the "indirect experience of illness" that caregivers endure. Similar to patients, caregivers experience the emotional stages of chronic illness, such as denial, anger, bargaining, depression, and acceptance [32]. This empathic connection creates a synchronization of emotions between the caregiver and the patient, intensifying the caregiver’s emotional burden [33,34]. Caregivers not only manage the logistical challenges of care but also share in the patient’s emotional journey, experiencing complex feelings like guilt, anger, and uncertainty [16,35]. This profound emotional connection underscores the need to support caregivers in regulating their emotions and developing effective coping strategies to navigate these challenges [35].
Caregiver Resources and Adaptation
The study also emphasized the dual nature of caregiver resources. Some caregivers reported sacrificing their own interests and well-being to focus entirely on the patient, often leading to exhaustion and depletion of inner resources. This phenomenon appears linked to cultural values of self-sacrifice, particularly in Italian culture, and disproportionately affects women [20]. Conversely, other caregivers demonstrated resilience by practicing self-care, engaging in activities such as exercise or taking time for themselves, which helped them preserve their internal resources and manage caregiving challenges effectively [33]. These findings highlight the importance of promoting self-care and resilience-building among caregivers.
Family Dynamics and Support Systems
The participants reported significant disruptions in family roles and routines due to caregiving, with primary caregivers often taking on responsibilities akin to "becoming their parent’s parent." This role reversal increased stress, responsibility, and emotional involvement. Beyond the primary caregiver, the entire family experienced changes in roles and relationships, necessitating support for all members. Previous research [38] suggests that families with strong internal adaptability can better manage the challenges of chronic illness. Additionally, recent systematic reviews emphasize the buffering effect of social support and professional guidance in mitigating the social impact of illness on patients, caregivers, and their families [39]. These findings underline the importance of addressing family systems as a unit of analysis and intervention in caregiving contexts.
In conclusion, this study reinforces the need for holistic approaches to caregiver support, emphasizing gender-sensitive interventions, resources to build resilience, and strategies to address the broader impact of caregiving on family systems. Such efforts are essential to mitigate the physical, emotional, and social toll of caregiving for elderly patients with chronic illnesses.
This could still be improved and the discussion expanded on.
Comments on the Quality of English LanguageDo review the entire paper for readability in English.
Author Response
Please see the file in the attachment
